

# Evaluation of predictive models for post-fire debris flows occurrence in the western United States

Efthymios I. Nikolopoulos[1], Elisa Destro[2], Md Abul Ehsan Bhuiyan[1], Marco Borga[2], Emmanouil N. Anagnostou[1]

[1] Department of Civil and Environmental Engineering, University of Connecticut, Storrs, CT, USA

[2] Department of Leaf, Environment, Agriculture and Forestry, University of Padova, Legnaro, PD, Italy

*Correspondence to*: Efthymios I. Nikolopoulos (efthymios.nikolopoulos@uconn.edu)

**Abstract.** Rainfall-induced debris flows in recently burned mountainous areas cause significant economic losses and human casualties. Currently, prediction of post-fire debris flows is widely based on the use of power-law thresholds and logistic regression models. While these procedures have served with certain success in existing operational warning systems, in this study we investigate the potential to improve the efficiency of current predictive models with machine-learning approaches. Specifically, the performance of a new predictive model based on random forest algorithm is compared against current techniques for the prediction of post-fire debris flow occurrence in the western United States. The analysis is based on a database on post-fire debris flows recently published by United States Geological Survey. Results show that predictive models based on random forest exhibit systematic and considerably improved performance with respect to the other models examined. In addition, the random forest-based models demonstrated improvement in performance with increasing training sample size, indicating a clear advantage regarding their ability to successfully assimilate new information. Complexity, in terms of variables required for developing the predictive models, deems important but the choice of model used is shown to have a greater impact on the overall performance.



## 1. Introduction

Wildfires constitute a natural hazard with devastating consequences to natural and built environment. In addition to the immediate impact of wildfire events to human lives, infrastructure and the environment, their adverse effects on landscape characteristics

generate a cascade of hydrogeomorphic hazards (Diakakis et al. 2017). One of the most frequent post-fire hazards is debris flow. Debris flows are rapidly flowing, gravity-driven mixtures of sediment and water commonly including gravel and boulders (Iverson, 2005), which rush down on steep channels and discharge onto debris fans, posing significant threat to downstream populations.

Post-fire debris flows (hereinafter DF) are predominantly derived from channel erosion and incision, usually generated during heavy precipitation events on burned areas (Cannon and DeGraff, 2009). Recent studies have shown that in fire affected regions the threat associated to debris flows may persist for several years after the fire incident (DeGraf et al., 2015; Diakakis et al., 2017), demonstrating the necessity for developing short and long-

term plans for the mitigation of this hazard (DeGraf et al., 2015).

In the western United States, DF is a well-recognized hazard that has claimed human lives and caused severe damages to infrastructure over the years (Cannon and Gartner, 2005). The occurrence of DF hazard in this region is expected to further intensify due to expected increase in fire occurrence and fire season length, as a result of climate change (Riley and

Loehman, 2016), and the continuous population growth on the wildland-urban interface (Cannon and DeGraff, 2009). Therefore, developing effective measures to reduce vulnerability of local communities to DF is of paramount importance.

Early warning is a critical element for the successful mitigation of DF hazard. Over the last decade a number of researchers have worked on developing procedures for predicting DF

in western US (Cannon et al., 2008; Cannon et al., 2011; Staley et al., 2013). In addition, federal agencies associated with monitoring and forecasting of natural hazards like the United States Geological Survey (USGS), National Oceanic and Atmospheric Administration (NOAA) and National Weather Service (NWS) have jointly developed a debris flow warning system for recently burned areas (Restrepo et al., 2008). In their vast

majority, the foundation of these warning procedures lies on empirical relationships that





are used to identify the conditions likely to lead to the occurrence of DF. In their simplest form, these relationships refer to rainfall intensity (or accumulation)-duration thresholds above which DF is likely to occur (Cannon et al., 2008; Cannon et al., 2011; Staley et al., 2013). Other procedures involve application of statistical models that incorporate

information on land surface characteristics (e.g. percentage of burned area, local topographic gradients etc.), in addition to rainfall properties, to predict the likelihood of a DF occurrence. The most commonly used statistical model for DF prediction is the logistic regression (Rupert et al., 2008; Cannon et al., 2010). Updates of these past prediction models were recently suggested by Staley et al. (2017), who proposed a new logistic

regression model that improves current DF prediction procedures in western United States. Additionally, in a recent study by Kern et al. (2017), a number of machine learning approaches were evaluated for DF prediction. The conclusions based on that study is that advanced statistical modeling techniques can offer significant improvement in the performance of current DF prediction models.

Both of the recent works of Staley et al. (2017) and Kern et al. (2017) suggest that although models for DF prediction may already exist for specific regions (Cannon et al., 2010), the importance of improving their accuracy and also extending prediction beyond the boundaries of these regions calls for continuous advancements of currently established procedures. Following this line of thought, this study focuses on the development of a new

DF prediction model that is based on a non-parametric statistical approach and the evaluation of its performance against state-of-art approaches for DF prediction in the western United States. Specifically, we evaluate the performance of four models that include i) rainfall accumulation-duration thresholds, ii) the logistic regression model suggested by Staley et al. (2017) and iii) two models based on the random forest technique

(Breiman, 2001) that are introduced in this study. In addition to the consistent evaluation of the performance of each model, this work investigates the relationship between prediction accuracy with complexity and data requirements (in terms of both record length and variables required) of each model. These are important aspects for selecting the most appropriate method and for providing guidance for data scarce regions at global scale.



## 2. Study area and data

This study is based on a USGS database that was recently published (Staley et al., 2016) and includes information on the hydrologic response of several burned areas in the western United States (Fig.1). The database reports the occurrence of debris flow (DF) or no-debris

flow (noDF), and rainfall characteristics for 1550 rainfall events in the period 2000-2012 together with field-verified information characterizing the areas affected by wildfires (Table 1). The area of fire-affected catchments analyzed varied between 0.02 and 7.9 km$^2$. Rainfall data were collected from rain gauges located within a maximum distance of 4 km from the documented response location. Reported rainfall characteristics included rainfall

peak intensities (and accumulations) at 15, 30 and 60 min time intervals, event total accumulation, duration and average intensity. Land surface characteristics of burned areas were recorded in order to evaluate the influence of the burned area to the hydrologic response. The differenced normalized burn ratio (dNBR), which is used for classification of burn severity (Miller and Thode, 2007; Keeley, 2009) is reported in addition with the

proportion of upslope catchment area at high or moderate severity and slope higher than 23°. Finally, since in burned areas changes in recovery vegetation increase erosion, the average erodibility index (KF-Factor) was recorded. The soil erodibility index gives evidence of erodibility and runoff potential of soil, taking into account fine fragments, organic fraction and soil texture of the soil. For more information on the estimation

procedures of the variables (Table 1), the interested reader is referred to Staley et al. (2016) and references therein.

The 334 events (~22%) of hydrologic responses in the database were identified as debris flows. The location of events in the dataset correspond predominantly to the area of southern California, which includes 61% (60%) of all records (DF records). Colorado

corresponds to 20% (10%) of the data (DF) and the rest of data correspond to other regions (Arizona, New Mexico, Utah, Idaho) of western US (Fig. 1). Since values for some of the variables (e.g. rainfall duration, 15min peak intensity etc.) are not reported consistently for all records, the analysis presented hereinafter is focused only on the 1091 events with complete record that involve the areas of Arizona (AZ), California (CA), Colorado (CO)

and New Mexico (NM).



## 3. Seasonality and characteristics of rainfall events

Most of the western United States is characterized by dry summers, when the fire activity is widespread, with a high percentage (50–80%) of annual precipitation falling during October-March. However, there are also regions, such as Arizona and New Mexico where heavy rains occur between July and August as a result of the North America monsoon (Westerling et al., 2006). More specifically, four different seasonal rainfall types characterize the Southwest United States (Moody and Marty, 2009): Arizona, Pacific, Sub-pacific, and Plains types. Arizona is characterized by dry spring, fall season is moist and winter and summer are wet; California is mainly characterized by Pacific type when maximum rainfall occur in winter season and summer is extremely dry. The Sub-pacific type, with wet winter, moist spring and summer, and fall dry, characterizes the southern part of Sierra Nevada region, a small area in the southern California. Climate similar to Arizona type characterizes the southwest Colorado, the East Colorado is characterized by Plains type, where the rainfall maxima occurs in summer. The Arizona type characterizes also the western New Mexico while the eastern part is characterized as Plain type.

Examination of the seasonality of the rainfall events analyzed (Table 2) demonstrates the similarities and differences attributed to the different climate types described above. The vast majority (92%) of rainfall events in Arizona occurred during the summer months (July and August). Similarly, the majority of rainfall events in western Colorado, occurred during late summer-early fall months (August and September). California, which is influenced by the Pacific rainfall regime, is dominated by winter rainfall events where 82% of events occurred between December and January. Seasonality of rainfall events for the New Mexico State exhibit a characteristic dominance of occurrences in summer with 94% of the events occurring in July and the remaining in June.

The North America monsoon is responsible for the summer rainstorms in these regions that typically last between June and mid-September, causing strong thunderstorm activities in the uplands of Arizona and New Mexico and the absent of rainy events in southern California (Mock, 1996; Adams and Comrie, 1997). September is the rainiest month in Colorado because of mid-latitude cyclones coming from Gulf of Alaska (Mock, 1996).



Differences in seasonality and large scale climatic controls, correspond essentially to differences in dominant precipitation type (e.g. convective vs stratiform) and differences in characteristic properties of rainfall events triggering debris flows (Nikolopoulos et al., 2015). Analysis of the characteristics of the rainfall events revealed clear regional dependences and for certain regions there were also distinct differences in the characteristics between DF and noDF events (Fig. 2).

Rainfall duration for events in Arizona and New Mexico is significantly lower than events in other regions, with California being associated with the longest duration events (10-70h in most cases), typical for the winter type rainfall that is dominant in this region. The DF-triggering events for Arizona, Colorado and New Mexico correspond to the lowest duration events while the opposite is shown for California (Fig.2a). Variability among regions and within noDF and DF-triggering events exists also for the magnitude of rainfall events (Fig.2b,c). With the exception of events in New Mexico, the other regions exhibit a distinct separation in the distribution of total rainfall accumulation (Fig.2b) and peak 15min accumulation (Fig.2c) between DF and noDF events. For these regions, the highest values for both variables are associated with the DF-triggering events, which justifies the rational for using these variables for predicting DF occurrence.

In addition to the marginal distribution of the rainfall variables shown in Figure 2, the relationship between duration and magnitude is presented in Figure 3. California events are distinctly clustered over the high duration-accumulation area (Fig.3a) demonstrating the already discussed regional dependence of rainfall characteristics. The total rainfall accumulation is strongly correlated with duration for the DF-triggering events (Pearson's correlation coefficient 0.7). On the other hand, the peak 15min accumulation, which is a proxy for max intensity of the events, does not correlate well with duration (Pearson's correlation coefficient -0.2). Overall, it is apparent from Figure 3 that there are areas in the accumulation-duration spectrum where the DF and noDF events are well mixed, which highlights the challenge of identification between the two and the need for classification approaches based on additional parameters.

Findings from the analysis of rainfall seasonality provide clear indications that there are distinct regional differences of the triggering rainfall characteristics. This justifies the development of regional predictive models as stated in past studies and raises an important



point of consideration for creating a single multi-region wide framework for DF prediction. The issue of regional dependence and how it can be incorporated into a single model is further discussed in section 4.1.3 below.

## 4. Methods

### 4.1 Models for predicting post-fire debris flow occurrence

This section describes the different models that will be evaluated for predicting post-fire debris flow occurrence. Selection of the different models is based on criteria of model simplicity, data requirements and relevance to common practice.

#### 4.1.1 Rainfall thresholds

Rainfall thresholds correspond to one of the simplest and most widely used approaches for predicting the occurrence of rainfall-induced mass movements such as shallow landslides and debris flows (Caine, 1980, Guzzetti et al., 2007, Cannon et al., 2011). Rainfall thresholds are commonly formulated as power-law relationships that link rainfall magnitude and duration characteristics as in the following:

$$E = \alpha D^{\beta} \tag{1}$$

where total event rainfall accumulation ($E$) is related to event duration ($D$). The intercept ($\alpha$) and exponent ($\beta$) are parameters estimated from the available observations. In this case, the threshold (hereinafter ED threshold) provides the rainfall accumulation above which a debris flow event will occur for a given duration. In this work, parameter $\beta$ was estimated according to the slope of $\log(E)$ vs $\log(D)$ using least squares linear regression and considering only the events that resulted in debris flows. The full record (both debris and no debris flow events) was then used to identify the optimum value of parameter $\alpha$. Details on the optimization of parameter $\alpha$ and the criteria used are discussed in section 4.4.

#### 4.1.2 Logistic regression

Another model that is frequently used for modeling the statistical likelihood of a binary response variable is the logistic regression (LR) model. In western United States, LR models were first developed for DF prediction almost a decade ago (Rupert et al., 2008, Cannon et al., 2010) and are still used to date (Staley et al., 2016, 2017).



The probability of occurrence ($P$) of PFDF according to logistic regression is given as

$$P = \frac{e^{x}}{1+e^{x}} \tag{2}$$

where the link function $x$ is modeled as a linear combination of one or more explanatory variables according to:

$$x = \gamma + \delta_1 X_1 + \delta_2 X_2 + .. + \delta_n X_n \tag{3}$$

where $X_n$ is the $n_{th}$ explanatory variable and $\gamma$ and $\delta_n$ are parameters estimated from the observation dataset. Selection of the explanatory variables is very crucial for successfully developing LR models. In this study, we adopted the latest LR model proposed by Staley et al. (2016, 2017), which can also be considered as the state-of-the-art for DF prediction in western US. After a thorough examination of several LR models, the authors of those works concluded that the most appropriate set of explanatory variables are:

$X_1$= max 15 min rainfall accumulation * proportion of upslope area burned at high or moderate severity with gradients $\geq 23°$

$X_2$= max 15 min rainfall accumulation * average dNBR normalized by 1000

$X_3$= max 15 min rainfall accumulation * soil KF-Factor

Parameters $\gamma$ and $\delta_n$ were estimated based on least squares regression. Specifically, the "glmfit" function of MATLAB software (version 2017b) was used to fit the binomial distribution to available data using the logit link function.

### 4.1.3 Random forest

Random forest (RF) is a non-parametric statistical technique that is based on decision tree ensemble (i.e. forest) procedure for classification or regression (Breiman, 2001). Despite being a well-known algorithm with extensive use in other fields (e.g. medicine); there are not many examples of RF applications in hydrogeomorphic response studies and most of them deal with landslide susceptibility (e.g. Brenning, 2005; Vorpahl et al., 2012; Catani et al., 2013; Trigila, 2015). Some of the main advantages of RF algorithm is that it allows to mix numerical and categorical variables and it does not require any knowledge on the distribution of variables and the relationship between them. In this work, we used RStudio software and the R package "randomForest" (Liaw and Wiener, 2002) to develop the RF model for PFDF prediction.





For the selection of the most important variables for the RF model we tested several different scenarios of variable combinations. During that investigation, we found that the use of an extra categorical variable (named "Region class" hereinafter) that is used to classify the dataset into two geographic regions (i.e. within California and other) improves

RF model performance and thus was included in the variables used for the RF development. Explanation for the importance of this regional distinction lies on the existence of a clear difference in the seasonality and subsequently rainfall characteristics between California and other regions considered. From all the different combinations of variables tested (results not shown) we identified two different models that we present and discuss in the

work. The first model (RF-ED) was developed using the variables of rainfall accumulation and duration. It is the model that we consider as the one with minimum data requirements, given that only two rainfall variables are used for the prediction. The second model (RF-all), is considered as the "data demanding" RF model and uses almost all available information on rainfall characteristics, burn severity, land surface properties etc. Table 3

reports all the variables used in each model.

### 4.2  Model performance criteria

Evaluation of model performance in predicting DF occurrence was based on the contingency table (Table 4), which is used to measure the number of correct/false predictions. True positive (*TP*) corresponds to the number of debris flow events correctly

predicted by the model, false positive (*FP*) indicates the number of falsely predicted debris flows, false negative (*FN*) is the number of missed debris flow events and true negative (*TN*) corresponds to the "NO debris flow" events correctly predicted. The metrics, according to the contingency table, that we use in the evaluation of the predictive skill of the models are the threat score (*TS*), the true positive rate (*TPR*) and the false positive rate

*(FPR)* defined as

$$TS = \frac{TP}{TP+FN+FP} \tag{4}$$

$$TPR = \frac{TP}{TP+FN} \tag{5}$$





$$FPR = \frac{FP}{TN+FP} \tag{6}$$

The threat score (known also as critical success index) provides information on the overall skill in predicting positive (i.e. DF) responses with respect to total (*TP+FP*) and missed (*FN*) positive predictions. *TPR* and *FPR* provide information on correct positive and false positive predictions as percentage of the total positive and negative events respectively. Lastly, the predictive performance of the different models examined is assessed based on

the Receiver Operating Characteristic (ROC) curves (Fawcett, 2006).

### 4.3   Identification of thresholds

Whether using ED, LR or RF models, identification of debris flow occurrence is based on the use of a threshold value above which we consider that a debris flow will occur. In the case of ED thresholds, the slope (parameter *β*) is estimated from the data (as discussed in

section 4.1.1) and the intercept (parameter *α*) is identified according to the maximization of *TS*. In other words, given the estimated parameter *β*, ED threshold is always defined in order to achieve maximum *TS* value for the dataset used to "train" the model.

LR and RF models estimate a probability of DF occurrence for both DF and NO DF events. Equivalently, this requires the selection of the appropriate threshold of probability value

above which we consider DF occurrence. Often the probability threshold corresponds to a value of 0.5 (see for example Staley et al., 2017) but this does not necessarily imply optimum performance considering that DF and NO DF events are not perfectly separated and some overlap in probability space exists (see example in Fig. 4). During the "training" of both the LR and RF models, we allow the probability threshold value to be defined

according, again, to the maximization of *TS* value.

### 4.4   Model training and validation framework

For training and validation of the predictive models we followed two different approaches that included a Monte Carlo random sampling and hold-one-out validation framework.



In the random sampling framework, a training dataset of size $M$ and a test dataset of size $K$ is sampled randomly from the original data sample. The training data are used to train each model (i.e. estimate parameters for ED and LR and build RF) and then the trained models are evaluated using the test data. This random sampling training-validation procedure was repeated 500 times to provide an estimate of the effect of sampling uncertainty on the model performance. The only condition that was imposed during the construction of the random train/test samples was the proportion of DF/NO DF events in each sample. We set the percentage of DF events to be 20% in both train and test following approximately the same percentage used in the training dataset of Staley et al., (2017), which corresponds roughly also to the percentage of DF events in the original sample as well. The test sample size $K$, was fixed to 100 while the train sample size $M$ was allowed to vary from 100 up to 900 to allow also investigation of the sensitivity of results to different train sample sizes.

In the hold-one-out validation, all events in the database except one, are used each time as the training dataset and the models are evaluated for each event that is left out. This procedure is repeated by sequentially holding out all events essentially allowing to validate the models against all available events. This process somewhat mimics better what would have been done in practice, considering that in an operational-like environment we would be training our predictive models with all available "historic events" and use them to predict the next "new" event. Therefore, in this case the training sample size was equal to 1090 and was constructed by sequentially leaving one event out from the original sample size. The training/validation process was repeated until all events were included as validation points (i.e. 1091 times).

## 5. Results and Discussion

In this section, we present and discuss the findings based on the evaluation results for the different predictive models and the two validation frameworks considered.

### 5.1 Random sampling validation

The random sampling validation results (Fig. 5) demonstrate the relative performance of the models examined as a function of the training sample size. Interestingly, even for the



smallest sample size examined (*M*=100) the RF-based models exhibit higher median values than the ED and LR models but at the same time are characterized by greater variability in their performance, manifested on the graph as larger boxes. As sample size increases the model performance (in terms of *TS* values) increases for both RF-based models. An

5 interesting point to note from these results is that for the smaller sample sizes examined (*M*=100-500) the RF-ED performed marginally better than the RF-all but as the sample size increased, the situation is reversed and higher *TS* values are associated with the RF-all model. This suggests that the increasing amount of data used for training improves at a higher rate the RF-based model that involves a greater number of explanatory variables.

On the other hand, both ED and LR models exhibit overall consistently lower performance than RF-based models and lower sensitivity to sample size.

A summary of the change of *TS* distribution with sample size is presented in Table 5, were the relative difference in median and interquartile range (IQR), between maximum (*M*=900) and minimum (*M*=100) sample size examined, is reported for each model. The

15 performance of RF-ED model improved significantly with an increase in the median value by 51 % and a decrease in the IQR by 20 %. The RF-all model showed even higher increase in the median (63 %) but variability in performance remained practically unchanged. On the other hand the ED and LR model showed much lower (than RF-based) increase in median (7 and 11 % respectively) but exhibited considerable decrease in IQR (25 % and

20 19 % respectively).

Furthermore, the relative difference between the models is presented for the highest sample size (*M*=900) and with reference to the LR model, which corresponds to the current state-of-the-art in PFDF occurrence prediction for western United States (Staley at al., 2017). Based on the results (Table 6) the RF-ED model *TS* is 55 % higher than the LR model but

with an increased IQR (+12 %). The RF-all model TS is 58 % higher but with significantly increased variability in performance (IQR +54 %). The ED model performs slightly lower ( -3 % in median) but with reduced variability (IQR -17 %) relative to the LR model.

The results from this random sampling validation exercise demonstrate a superior performance of the RF-based models particularly for the largest sample sizes examined.

For the smallest sample size, the RF-based models are characterized by significant variability in their performance, which may raise questions regarding their applicability.



However, for higher sample sizes and despite the fact that variability remains, the median performance increases to the degree that makes clear the distinction in performance with respect to other models.

## 5.2 Hold-one-out validation

For the hold-one-out validation, results are reported by considering collectively the model prediction outcome for all events, meaning that the prediction of all 1091 events were used to summarize the performance indicators (*TS*, *TPR*, *FPR*) reported in Table 7. Recall that for the prediction of each event, each model was trained with the remaining dataset (i.e. 1090 events) and thresholds were determined according to the maximization of *TS* in each case. According to the *TS* values reported, the RF-based models with *TS* equal to 0.63 (RF-ED) and 0.64 (RF-all) exhibit considerably improved performance with respect to the ED and LR models with *TS* values of 0.41 for both models. Comparison of the *TPR* and *FPR* values suggests that the superiority of RF-based models is primarily attributed to the lower false alarm rates (≤ 11 %) relative to ED and LR models (~30 %). The true positive rate appears equivalent among the different models.

However, an important note here is that these metrics (*TS*, *TPR*, *FPR*) depend highly on the selection of the threshold. So far in the analysis we have considered the identification of thresholds based on maximization of *TS*. To further investigate the dependence of results for varying thresholds we evaluated the model performance considering a variable threshold and reported the results based on the Receiver Operating Characteristic (ROC) curves (Fawcett, 2006). In the ROC graph (Fig. 6), the point (0, 1), which corresponds to 100 % *TPR* and 0 % *FPR* represents the points of perfect prediction. The 45º line correspond to a random predictor (i.e. 50 % of the times being correct) and any point above that line corresponds to a model with some predictive skill. The ROC curves demonstrate the model's predictive performance for different thresholds and the higher the area under the curve (AUC) the more skillful the model is. From a visual examination of the ROC curves in Fig. 6, one can quickly identify a number of main points regarding the predictive skill of the models examined in this study. First, all models show significant skill (i.e. large departure from 45º line). Second, the performance of all models is highly dependent on




the selection of the threshold. Third, the performance for the thresholds corresponding to maximization of *TS* (denoted as solid circles in Fig. 6) does not necessarily coincide with the point of best available performance (i.e. point closer to point (0,1)). The ROC curves from RF-based models demonstrate once again the superior performance of both RF

models examined while the ROC curves from ED and LR models are relatively close. Based on the corresponding AUC value for each model, which provides a mean of quantification for the comparison of their performance, we can rank the models in increasing performance as follows: 0.77 (ED), 0.80 (LR), 0.90 (RF-ED) and 0.94 (RF-all). Based on these results, the choice of LR is justified relative to the use of a simple power-

law ED model, but it still remains inferior to the RF-based models for all threshold values examined. Comparison between RF-ED and ED models highlights the benefit of using a machine learning approach in predictive modeling. Considering that both these models are developed based on the same information (rainfall accumulation and duration), it is noteworthy how the technique involved random forest (in contrast to power law threshold)

can impact the respective performance.

## 6.  Conclusions

In this study, we evaluated the performance of four different models for post-fire debris flow prediction in western United States. The analysis was based on a dataset that was recently made available by USGS and the models involved included the current state-of-

the-art, which is a recently developed model based on logistic regression, a model based on rainfall accumulation-duration thresholds, considered as the simplest one followed in practice worldwide, and two models based on the random forest algorithm that were developed in this study. We investigated the relationship between prediction accuracy with model complexity and data requirements (in terms of both record length and variables

required) of each model. According to the results from this analysis we found that the application of random forest technique leads to a predictive model with considerably improved accuracy in the prediction of post-fire debris flow events. This was attributed mainly to the ability of RF-based models to report lower values of false alarm rates. The currently used LR model performed better that the simple ED model, but it was



outperformed by both RF-based models, particularly as the training sample size increased. Increasing sample size has a profound effect on improving the median performance of RF-based models while variability of the performance remained significant for all sample sizes examined. On the other hand, the LR and ED models exhibited minimal improvement in

the median performance but more considerable reduction in the IQR with increasing sample size. Comparison between the two RF-based models suggests that even the model with significantly less data requirements (i.e. RF-ED) constitutes a relatively good predictor. Overall the more "complex" model (RF-all) exhibited the best performance. Analysis of sample size sensitivity showed that increasing data variables can lead to

increasing performance, but, this comes at a cost on data availability for properly training the more "data-demanding" models. Lastly, the ROC analysis indicated that the performance of the various predictive models is highly related to the selection of thresholds. Selection of thresholds should be based on operator/stakeholder criteria who can identify the threshold according to the target *TPR* and tolerance at *FPR* of the prediction

system at hand.

Future work will be focused on examining the model performance using alternative sources of rainfall information such as quantitative precipitation estimates from weather radar, satellite-based sensors and numerical weather prediction models (e.g. ensemble weather forecasts). In conclusion, although current findings provide a clear indication that random

forest technique improves prediction of post-fire debris flow events, it is important to note that there may be other approaches (see for example Kern et al., 2017) that can offer additional advantages, therefore future investigations should expand also on the investigation of other machine learning or statistical approaches for developing post-fire debris flow prediction models.

## Acknowledgements

This work was funded by the Eversource Energy Center at the University of Connecticut (grant number 6204040). The authors would like to sincerely acknowledge the efforts of United States Geological Survey for compiling and making available a unique dataset that

made this research work possible. Special thanks go to Dr. Jason Kean of USGS for providing information on the dataset.





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





**Table 1 A summary of variables reported in the post-fire debris flow database.**

| Variables | Unit of measurement |
|---|---|
| Hydrological response | (DF=1,NO DF=0) |
| Storm Duration | h |
| Storm Accumulation | mm |
| Average storm intensity | $mmh^{-1}$ |
| Peak 15-minute rainfall intensity (accumulation) | $mmh^{-1}$ (mm) |
| Peak 30-minute rainfall intensity (accumulation) | $mmh^{-1}$ (mm) |
| Peak 60-minute rainfall intensity (accumulation) | $mmh^{-1}$ (mm) |
| Contributing Area | $km^2$ |
| Proportion of upslope area with moderate or high dNBR and slope higher than 23° | % |
| Average differenced normalized burn ratio (dNBR) /1000 | - |
| Average KF Factor (soil erodibility index) | - |



**Table 2 Monthly distribution of rainfall events analyzed for Arizona, California, Colorado and New Mexico. Values in table correspond to percentage (%) of total number of events analyzed per region.**

|     | Jan | Feb | Mar | Apr | May | Jun | Jul | Aug | Sep | Oct | Nov | Dec |
|-----|-----|-----|-----|-----|-----|-----|-----|-----|-----|-----|-----|-----|
| AZ  |     |     |     |     |     |     | 54  | 38  | 8   |     |     |     |
| CA  | 42  | 10  | 2   |     |     |     |     |     |     | 1   | 5   | 40  |
| CO  |     |     |     |     |     | 5   | 2   | 24  | 68  | 1   |     |     |
| NM  |     |     |     |     |     | 6   | 94  |     |     |     |     |     |



**Table 3 Description of variables included in the development of RF models. Symbol "X" denotes the variables that were included in each model.**

| Variables | RF models | |
|---|---|---|
| | RF-ED | RF-all |
| Region class | X | X |
| Rainfall accumulation (mm) | X | X |
| Rainfall duration (h) | X | X |
| Peak 15min rainfall (mm) | | X |
| Proportion of upslope area with moderate or high dNBR and slope higher than 23° | | X |
| Average differenced normalized burn ratio (dNBR) /1000 | | X |
| Average KF Factor (soil erodibility index) | | X |



**Table 4 Contingency table.**

|  |  | Observed | |
| --- | --- | --- | --- |
|  |  | Debris Flow | NO Debris Flow |
| **Predicted** | Debris Flow | *TP* | *FP* |
|  | No Debris Flow | *FN* | *TN* |





**Table 5 Relative Change (%) of *TS* distribution between 900 and 100 sample size.**

| Relative Change (%), between 100 and 900 sample size | | |
|---|---|---|
| | Median | IQR |
| RF-ED | +51 % | -20 % |
| RF-all | +63 % | 0.4 % |
| Power Law - ED | +7 % | -25 % |
| LR | +11 % | -19 % |





**Table 6 Relative difference (%) between the reference model (LR) and the RF-ED, RF-all and ED models at the maximum sample size examined. Positive values denote increase of other models relative to LR.**

| Relative Difference (%) reference to LR at 900 sample size | | |
|---|---|---|
| | Median | IQR |
| RF-ED | +55 % | +12 % |
| RF-all | +58 % | +54 % |
| Power Law - ED | -3 % | -17 % |
| LR | - | - |



**Table 7 Model performance according to the thresholds based on maximization of *TS*.**

|  | *TPR* | *FPR* | *TS* |
|---|---|---|---|
| RF-ED | 0.84 | 0.11 | 0.63 |
| RF-all | 0.76 | 0.06 | 0.64 |
| Power Law - ED | 0.78 | 0.31 | 0.41 |
| LR | 0.77 | 0.29 | 0.41 |



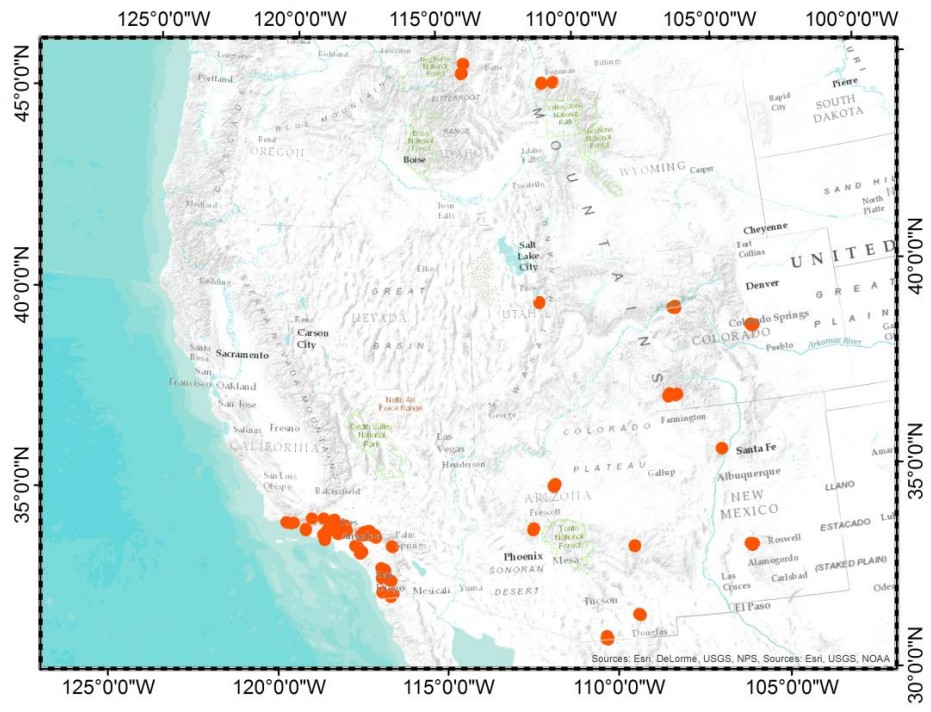

**Figure 1 Location of all post-fire debris flow records included in the USGS database. Note that all events in Utah and Idaho were excluded from the analysis due to their incomplete record of variables.**



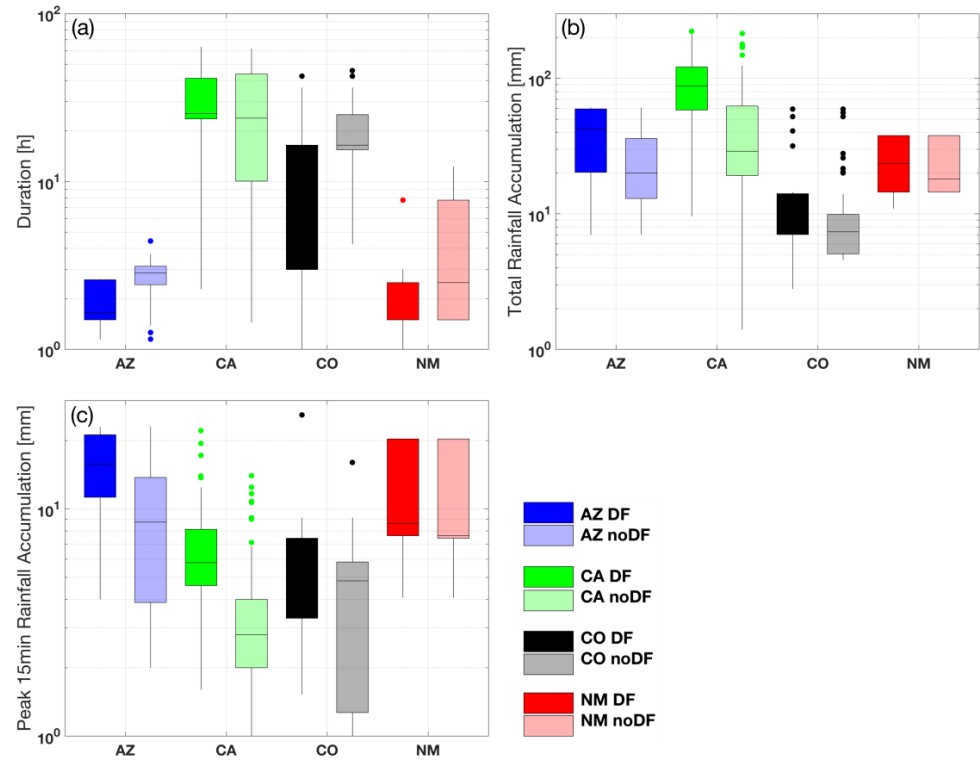

**Figure 2 Boxplot for storm duration, storm accumulation and peak 15-min storm accumulation for Arizona (blue), California (green), Colorado (black) and New Mexico (red). Dark (light) colors correspond to DF and noDF events.**



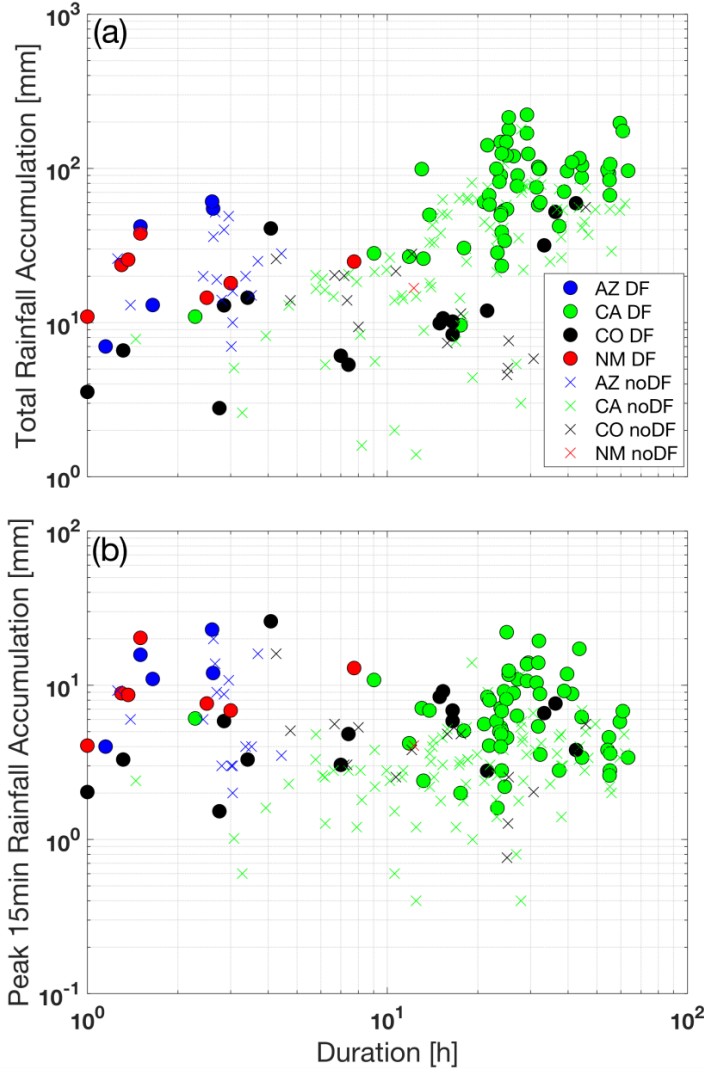

**Figure 3. Total rainfall accumulation versus duration (a) and peak 15 minute rainfall**

5    **accumulation versus duration (b) for Arizona, California , Colorado and New Mexico.**

**Colored dots and x symbols correspond to debris flow and no debris flow occurrence**

**respectively.**



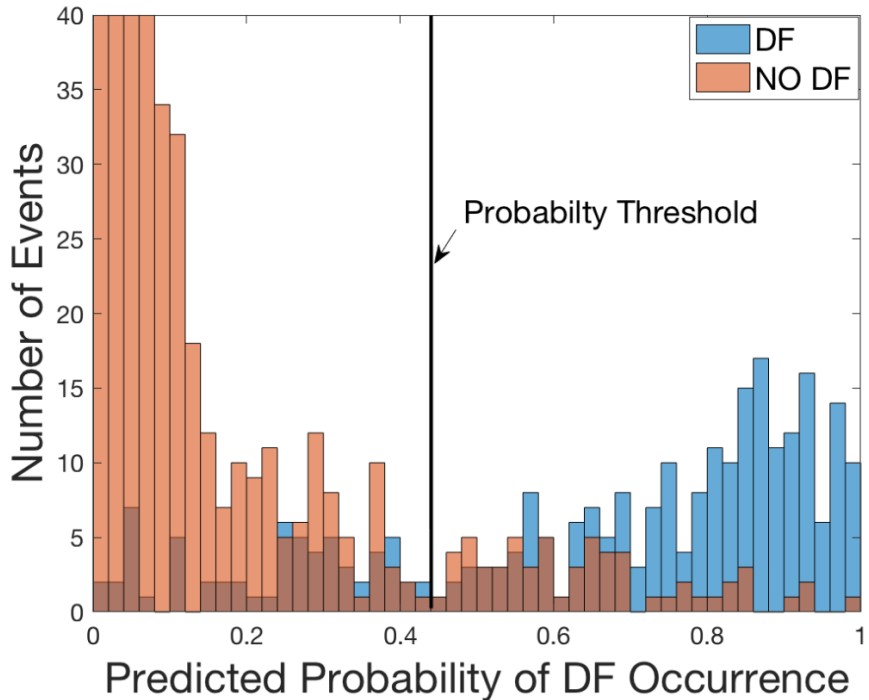

**Figure 4 Example plot demonstrating the predicted probability of debris flow occurrence for both DF and noDF events. The selected threshold corresponding to**
5 **optimum (i.e. max *TS*) performance is also shown.**



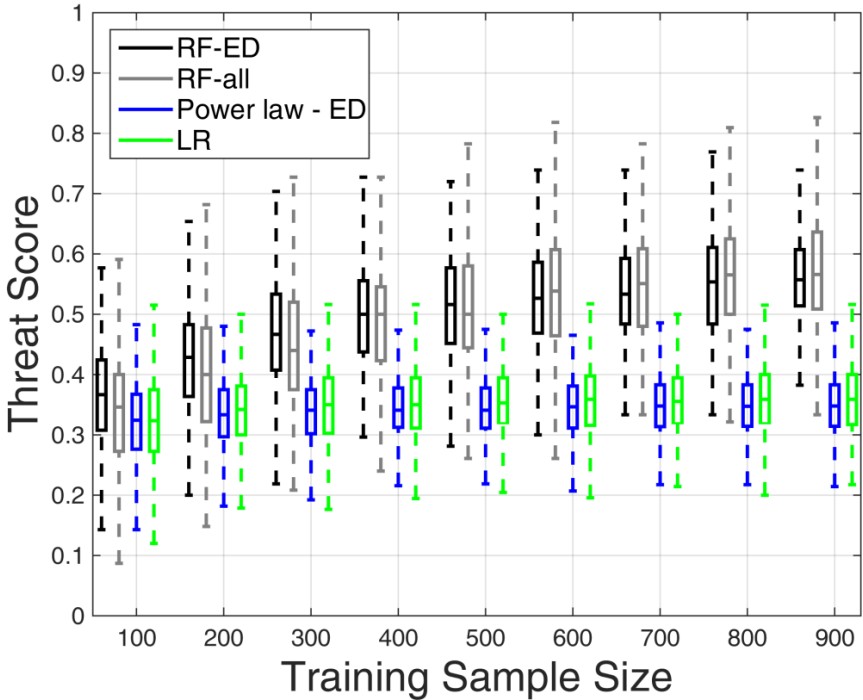

**Figure 5 Sensitivity to sample size: boxplots of the Threat Score (*TS*), according to**
5 **random sampling validation framework, for increasing sample size and for the 4**
**models considered for post-fire debris flow prediction.**





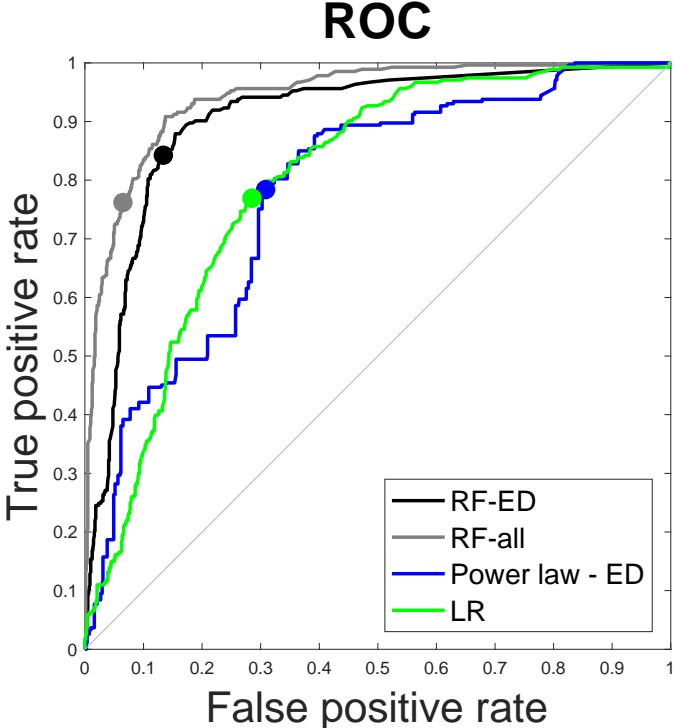

**Figure 6 ROC curves for the hold-one-out validation technique for the four models.**
**Circle symbols correspond to the model performance when selected thresholds were**
5   **based on *TS* maximization.**