# Peer review of "Evaluation of predictive models for post-fire debris flows occurrence in the western United States"

_Natural Hazards and Earth System Sciences, 2018_

## Referee Comment (RC1) · Anonymous Referee #1 · 10 May 2018

GENERAL COMMENTS

The contribution "Evaluation of predictive models for post-fire debris flows occurrence in the western United States" by Efthymios I. Nikolopoulos and co-Authors is good and potentially publishable. The Authors in this study we investigate the potential to improve the efficiency of current predictive models with machine-learning approaches. The analysis is based on a database on post-fire debris flows published by United States Geological Survey. In general, the topic well address scientific questions within the scope of NHESS.

The theoretical background is well-argued. Review of literature seems complete. The description of study area is not sufficiently complete. The description of methodology and successive parts of paper are well organized but can be improved. Results, and discussion sections are short compared to amount of work done. They should be widely increased. The readability of the whole paper is sufficient with a quite good English. Overall, the work presents some carelessness and incompleteness. It can be published on NHESS journal only after a major revision.

SPECIFIC COMMENTS

I have some specific comments that should be addressed before the manuscript can be accepted for publication.

☐        Section 1, "Introduction" (Page 2-3).

At Page 3, Lines 23 - 25 the Authors should add references about the rainfall accumulation-duration thresholds, and the two model used.

☐        Section 2, "Study area and data" (Page 4, Lines 1-30).

This paragraph is confuse and not complete. The Fig.1 does not show well the study area, the four different regions, the location of the DF, and the area of fire-affected catchments. In addition, the Authors should better explain the differenced normalized burn ratio (dNBR) used in your analysis (Page 4, Lines 13-14).
For a better clarity and understanding of the text, Authors should specify in detail how the values of the average erodibility index (KF-Factor) for the four different regions were chosen (Page 4, Lines 16-21).

☐        Section 3, "Seasonality and characteristics of rainfall events" (Page 5-7).

At the page 5, Lines 16-24 the Authors refer to statistics about the seasonality and characteristics of rainfall events reported in Table 2. The total number of the events is normalized in terms of percentage, this well describe the distribution of rainfall event analyzed for the different regions. However, this does not allow a comparison by number of events for each region. This comparison could be useful to guarantee the correctness of the considerations set out in Section 3 (Page 6, lines 4-28) and in Section 4.1.3 (Page 9, lines 6-8). I suggest to integrate in the Table 2 the absolute and relative frequency of the DF and noDf events.

☐        Section 4.1.2 "Logistic regression" (Page 7-8).

The LR model works with either continuous or categorical independent variables, or a combination of the two types, regardless whether they present a normal distribution or not (Costanzo et al., 2014). The underlying mathematical relationship between the dependent dichotomous variable (presence/absence of a landslide in the mapping unit and the $n$ independent variables ($X_1$, ...,$X_n$). Authors adopt the logistic regression (LR) model using a set of explanatory variables ($X_1$, $X_2$, and $X_3$) that have in common the cumulated rainfall (max 15 min rainfall accumulation). My question is: the explanatory variables used for the Authors are really independent? For a better clarity and understanding, Authors should specify in detail this aspect.

☐        Section 4.1.3 "Random forest" (Page 8-9).

Authors stated that "*The first model (RF-ED) was developed using the variables of rainfall accumulation and duration. It is the model that we consider as the one with minimum data requirements, given that only two rainfall variables are used for the prediction.*" (Page 9, Lines 10-12), but in the Page 9, (Lines 2-5) the Authors indicate they have used also an extra categorical variable (named "Region class"). The authors should better explain the total number of variables used and verify if it is in agreement with Table 3.

☐      Section 4.3 "Identification of thresholds" (Page 10, Lines 11-25).

This section is very short. Authors should better argue and comment the obtained results. I suggest to include in the paper a new Figure that show the distribution of the (D,E) pairs with the rainfall thresholds and their uncertainties. In addition the Authors should better explain how they reconstructed the rainfall conditions responsible of the DF and noDF. In particular, if have been used the subjective or objective methods (Vessia et al., 2014; Rossi et al. 2017; Melillo et al. 2015, 2018).

TECHNICAL CORRECTIONS

Page 1, Line 21: After Page 1, Line 20 it seems that there are two parts in the text that are not well connected. I suggest to rewrite better the start of this part.

Page 1, Line 24: I suggest to introduce here the acronyms of the four regions (AZ, CA,CO, and NM), because are here that are cited for the first time.

Page 7, Line 21: I suggest to replace "both debris and no debris flow events" with "both DF and noDF events"

Page 8, Line 1: The acronym "PFDF" does not exist. I suggest to introduce PFDF in the previous part (Page7, Line 7)

Page 8, Line 12, 14, 15: I suggest to replace "$X_1$, $X_2$, $X_3$" with "$X_1$, $X_2$, $X_3$".

Page 10, Line 18, 22: I suggest to replace "NO DF" with "noDF".

Page 11, Line 7: I suggest to replace "NO DF" with "noDF".

Page 19, Table 1: I suggest to replace "(DF=1,NO DF=0)" with "(DF=1, noDF=0)"

Page 22, Table 1: I suggest to replace "(NO Debris Flow)" with "no Debris Flow" and "(No Debris Flow)" with "no Debris Flow".

Page 26, Figure 1: Please show better in the Figure the study area, the four different regions, the location of the DF, and the area of fire-affected catchments. In addition, insert horizontal scale and North indicator symbol.

Page 27, Figure 2: In the blue (AZ region) and black (CO region) boxplots the median is not well visible. Please change the colours using other with major contrast. In addition I suggest to replace in the caption of the Figure "Figure 2 Boxplot for storm duration, storm accumulation and peak 15-min storm accumulation" with "Figure 2 Boxplot for (a) storm duration, (b) storm accumulation and (c) peak 15-min storm accumulation".

Page 28, Figure 3: I suggest to replace the caption with "(a) Total rainfall accumulation vs duration and (b) peak 15 minute rainfall accumulation vs duration for Arizona (AZ), California (CA), Colorado (CO) and New Mexico (NM). Colored dots and x symbols correspond to DF and noDF occurrence respectively".

Page 30, Figure 5: I suggest to use a different type of the line (solid line) to represent the full range of variation. The dashed-line produce confusing.

References:

Costanzo, D., Chacón, J., Conoscenti, C., Irigaray, C., and Rotigliano, E. (2018). Forward logistic regression for earth-flow landslide susceptibility assessment in the Platani river basin (southern Sicily, Italy), Landslides, 11, 639–653, https://doi.org/10.1007/s10346-013-0415-3, 2014.

Melillo, M., Brunetti, M.T., Peruccacci, S., Gariano, S.L., Guzzetti, F., (2015). An algorithm for the objective reconstruction of rainfall events responsible for landslides. Landslides 12(2), 311–320, https://doi.org/10.1007/s10346-014-0471-3

Melillo M., Brunetti M. T., Peruccacci S, Gariano S. L., Roccati A., Guzzetti F. (2018). A tool for the automatic calculation of rainfall thresholds for landslide occurrence. Environmental Modelling & Software, https://doi.org/10.1016/j.envsoft.2018.03.024

Rossi, M., Luciani, S., Valigi, D., Kirschbaum, D., Brunetti, M.T., Peruccacci, S., Guzzetti, F., 2017. Statistical approaches for the definition of landslide rainfall thresholds and their uncertainty using rain gauge and satellite data. Geomorphology 285, 16–27, https://dx.doi.org/10.1016/j.geomorph.2017.02.001

Vessia, G., Parise, M., Brunetti, M. T., Peruccacci, S., Rossi, M., Vennari, C., Guzzetti, F., 2014. Automated reconstruction of rainfall events responsible for shallow landslides. Nat. Hazards Earth Syst. Sci.14(9), 2399-2408, https://doi.org/10.5194/nhess-14-2399-2014.

---

## Referee Comment (RC2) · Anonymous Referee #2 · 21 May 2018

This paper is intended to present an improved model for forecasting debris-flow occurrence in recently burned areas. The presented material relies exclusively upon previously published data from other sources, and existing methodology. Given the relatively narrow target audience and lack of uniqueness of this topic, my criteria for recommending the publication of this type of study are as follows: 1) New insight into the mechanisms responsible for postwildfire debris-flow occurrence must be gained. 2) Unique methodology for improved predictions must be presented. 3) The methods used to compare model predictions must be defensible, and account for uncertainty in model predictions.

[Figure]

While this paper is well-written, significant revision is required in order to meet the specified criteria for the following reasons:

1) No new insight into the prediction of postwildfire debris-flow occurrence or the processes responsible for debris-flow generation have been presented within. The authors used an existing database of debris-flow occurrence, storm characteristics, and a select number of variables to test an existing modeling approach. The authors did not attempt to incorporate different metrics of topography, fire severity, or other physiographic properties of watersheds into this analysis. As such, the author's solely relied upon someone else's calculation of metrics and interpretations of the variables that are important for inclusion in this analysis. Hence, we did not learn anything about new factors or combinations of variables that may be relevant for the prediction of postwildfire debris-flow occurrence.

2) The authors present a method that utilizes an existing methodology (random forest analysis). It's unclear to me how or why this is significantly different, or a significant advancement, over the machine learning method presented by Kent et al., 2017. It seems that the comparison of model predictions focuses almost exclusively on those between the models presented by the author and the USGS logistic regression model.

3) The manuscript provides methodology for a binary classifier model used to predict the occurrence of a postwildfire debris flow. The authors compare their predictions to the USGS logistic regression model, but significantly simplify the USGS model predictions. Logistic regression analysis produces a continuous estimate of the likelihood of debris flow occurrence with values that range from 0 - 1, with 1 representing the highest likelihood. In this study, the authors consider the LR model as a binary classifier model, failing to consider that the model is not inherently binary. The analysis then proceeds to use ROC metrics for binary classifier models to compare the results. The Staley et al 2017 manuscript that the authors draw their comparative statistics from does use ROC metrics to compare findings for a portion of the results and discussion. In my opinion, this is a significant flaw in the Staley et al. 2017 manuscript.

By only viewing the USGS model as a binary classifier, the authors oversimplify the predictions made by the USGS logistic regression model by not accounting for the implicit uncertainty in the logistic regression model predictions. For example, the USGS model might estimate a 0.51 likelihood for a debris flow (i.e. just slightly better than a coin toss), but no debris flow might have occurred. This manuscript considers this to be a failed prediction, but is it really? I would argue that a model that estimated a 0.49 likelihood of not having a debris flow is not incorrect. Hence, the conclusion that the presented model significantly outperforms the USGS model predictions based upon the analysis of the model as a binary classifier is misleading.

––––––––––––––––––––––––––

---

## Author Comment (AC1) · 11 Jul 2018

Please see zip file uploaded as supplementary material.

Please also note the supplement to this comment:
https://www.nat-hazards-earth-syst-sci-discuss.net/nhess-2018-85/nhess-2018-85-AC1-supplement.zip
* * *

---

## Author Comment (AC2) · 11 Jul 2018

Please see our response uploaded as supplementary material.

Please also note the supplement to this comment:
https://www.nat-hazards-earth-syst-sci-discuss.net/nhess-2018-85/nhess-2018-85-AC2-supplement.zip
* * *

---

## Author Response (AR1)

**Authors response to comments by the reviewers**

Manuscript reference number: nhess-2018-85

Manuscript title: Evaluation of predictive models for post-fire debris flows occurrence in the western United States.

We would like to thank the reviewers for their constructive comments. We have revised the manuscript according to these comments and below we provide our detailed response to each of their comments. Comments from the reviewer are in black font and our response in blue font.

**Reviewer #1**

GENERAL COMMENTS

**1.1** The contribution "Evaluation of predictive models for post-fire debris flows occurrence in the western United States" by Efthymios I. Nikolopoulos and co-Authors is good and potentially publishable. The Authors in this study we investigate the potential to improve the efficiency of current predictive models with machine-learning approaches. The analysis is based on a database on post-fire debris flows published by United States Geological Survey. In general, the topic well address scientific questions within the scope of NHESS.

Response

We would like to thank the reviewer for his/her comment.

**1.2** The theoretical background is well-argued. Review of literature seems complete. The description of study area is not sufficiently complete. The description of methodology and successive parts of paper are well organized but can be improved. Results, and discussion sections are short compared to amount of work done. They should be widely increased. The readability of the whole paper is sufficient with a quite good English. Overall, the work presents some carelessness and incompleteness. It can be published on NHESS journal only after a major revision.

Response

We have revised the manuscript according to reviewer's general comments (expanding results and discussion section) and specific comments (see response to specific comments below).

SPECIFIC COMMENTS

**1.3** I have some specific comments that should be addressed before the manuscript can be accepted for publication.

Response

We have addressed in detail all specific comments (see response below and revised version of the manuscript).

**1.4** Section 1, "Introduction" (Page 2-3).

At Page 3, Lines 23 - 25 the Authors should add references about the rainfall accumulation-duration thresholds, and the two model used.

Response

We have added references for the rainfall accumulation-duration thresholds as follows

"…*include i) rainfall accumulation-duration thresholds (Guzzetti et al., 2007, Cannon et al., 2011; Rossi et al. 2017; Melillo et al. 2018),…*".

The two models (RF-ED and RF-all) were developed in this work, therefore at that point of the manuscript only the random forest algorithm is referenced.

**1.5** Section 2, "Study area and data" (Page 4, Lines 1-30).

This paragraph is confuse and not complete. The Fig.1 does not show well the study area, the four different regions, the location of the DF, and the area of fire-affected catchments. In addition, the Authors should better explain the differenced normalized burn ratio (dNBR) used in your analysis (Page 4, Lines 13-14). For a better clarity and understanding of the text, Authors should specify in detail how the values of the average erodibility index (KF-Factor) for the four different regions were chosen (Page 4, Lines 16-21).

Response

We have revised Figure 1, please see our response to comment 1.19 below.

We have revised text to include more info on dNBR as follows

*"Information on burn severity was based on the differenced normalized burn ratio (dNBR) (Key and Benson, 2006), calculated from near infrared and shortwave infrared observations, which is frequently used for classification of burn severity (Miller and Thode, 2007; Keeley, 2009). Severity classification from dNBR was validated from field observations provided by local burned area emergency response teams."*

For soil erodibility index we have updated text and references. Specifically, we provide reference to STATSGO database that was used to derive information on Kf factor and revised the text as follows

*"Finally, since in burned areas changes in recovery vegetation increase erosion, the average erodibility index (Kf factor) derived from the STATSGO database (Schwartz and Alexander, 1995) is reported in the database as well. Kf factor provides evidence of erodibility of soil, taking into account the fine-earth fraction (<2mm).)."*

**1.6** Section 3, "Seasonality and characteristics of rainfall events" (Page 5-7).

At the page 5, Lines 16-24 the Authors refer to statistics about the seasonality and characteristics of rainfall events reported in Table 2. The total number of the events is normalized in terms of percentage, this well describe the distribution of rainfall event analyzed for the different regions. However, this does not allow a comparison by number of events for each region. This comparison could be useful to guarantee the correctness of the considerations set out in Section 3 (Page 6, lines 4-28) and in Section 4.1.3 (Page 9, lines 6-8). I suggest to integrate in the Table 2 the absolute and relative frequency of the DF and noDf events.

Response

We have modified/updated Table 2 per reviewer's suggestion. The revised Table 2 now reports both the actual number of events and monthly distribution of both DF and noDF events.

**1.7** Section 4.1.2 "Logistic regression" (Page 7-8).

The LR model works with either continuous or categorical independent variables, or a combination of the two types, regardless whether they present a normal distribution or not (Costanzo et al., 2014). The underlying mathematical relationship between the dependent dichotomous variable (presence/absence of a landslide in the mapping unit and the $n$ independent variables ($X_1$, ...,$X_n$). Authors adopt the logistic regression (LR) model using a set of explanatory variables ($X_1$, $X_2$, and $X_3$) that have in common the cumulated rainfall (max 15 min rainfall accumulation). My question is: the explanatory variables used for the Authors are really independent? For a better clarity and understanding, Authors should specify in detail this aspect.

Response

Based on our interpretation of the reviewer's comment, we assume that he/she wonders whether inclusion of the max 15min rainfall accumulation in all explanatory variables affects the dependence between explanatory variables.

As it is shown in the original version of the manuscript, the three variables $X_1, X_2, X_3$ are defined as the product of max 15min rainfall accumulation with the 1)average normalized dNBR, 2) proportion of upslope area burned with gradients >23deg,and 3) soil KF-Factor. Therefore, the correlation (i.e. dependence) between those 3 variables (dNBR, upslope burned % and KF-factor) is not affected by the multiplication with the rainfall accumulation (i.e. correlation between x1 and x2 is the same as correlation between a*x1 and a*x2). Essentially, in the formulation of explanatory variables (proposed by Staley et al. 2016, 2017) the 15min rainfall accumulation is used to weigh the three variables mentioned above.

We clarify this in the revised version by stating that

"*Based on this formulation, information on the maximum 15 min rainfall accumulation is used to weigh the other three parameters (upslope burned area, average dNBR and KF-factor) considered.*"

**1.8** Section 4.1.3 "Random forest" (Page 8-9).

Authors stated that "*The first model (RF-ED) was developed using the variables of rainfall accumulation and duration. It is the model that we consider as the one with minimum data requirements, given that only two rainfall variables are used for the prediction.*" (Page 9, Lines 10-12), but in the Page 9, (Lines 2-5) the Authors indicate they have used also an extra categorical variable (named "Region class"). The authors should better explain the total number of variables used and verify if it is in agreement with Table 3.

Response

We would like to thank the reviewer for pointing this inconsistency. We have revised the text appropriately according to the following

"*The first model (RF-ED) was developed using the variables of rainfall accumulation, duration and region class. It is the model that we consider as the one with minimum data requirements, given that only two rainfall variables and a region classification is used for the prediction.*"

**1.9** Section 4.3 "Identification of thresholds" (Page 10, Lines 11-25).

This section is very short. Authors should better argue and comment the obtained results. I suggest to include in the paper a new Figure that show the distribution of the (D,E) pairs with the rainfall thresholds and their uncertainties. In addition the Authors should better explain how they reconstructed the rainfall conditions responsible of the DF and noDF. In particular, if have been used the subjective or objective methods (Vessia et al., 2014; Rossi et al. 2017; Melillo et al. 2015, 2018).

Response

We would like to note that section 4.3. belongs to "Methodology" section and as such, it is not the appropriate section to present results. In section 4.3. we describe the methodology for selecting the thresholds in E,D space (for ED model) or probability space (for LR and RF). To improve our description for the ED threshold we have updated Figure 4 to demonstrate an example of selected ED threshold that maximizes threat score.

Rainfall characteristics were calculated using the approach described in Kean et al. (2011). We have added the following in the manuscript to clarify this aspect.

*"According to the description of the dataset provided in Staley et al. (2016), the rainfall characteristics (peak intensities, accumulation etc) were calculated using a backwards differencing approach (Kean et al., 2011)."*

Uncertainty is a very important aspect and we would like to thank that reviewer for highlighting this. Although we do not explicitly present the uncertainty in the model parameters (e.g. for ED models) we present the variability in the model performance as a result of the sampling uncertainty during the random sampling validation. To better describe this and discuss in more detail the importance of uncertainty, we have revised the text in section 5.1 as follows:

*"Additionally, an important note is that overall the variability of the performance of all models, for a given sample size, is considerable and this essentially highlights the effect of sampling uncertainty; an aspect that requires careful consideration for the development and application of such predictive models. "*

and conclusions section as follows:

*"Uncertainty is a very important element to consider when developing and evaluating predictive models of this nature. Two important sources of uncertainty pertain to estimation of input variables (e.g. rainfall, burn severity) and sampling. In this work, we implicitly demonstrated the impact of sampling uncertainty on model's prediction skill through the random sampling exercise but we did not account for uncertainty in input parameters. The impact of input parameter uncertainty will be a topic of future research."*

**1.10** Page 1, Line 21: After Page 1, Line 20 it seems that there are two parts in the text that are not well connected. I suggest to rewrite better the start of this part.

Response

Page 1 refers to the abstract of the manuscript. We believe that rewriting the abstract is not required.

**1.11** Page 1, Line 24: I suggest to introduce here the acronyms of the four regions (AZ, CA,CO, and NM), because are here that are cited for the first time.

Response

We agree with the reviewer and we revised accordingly.

**1.12** Page 7, Line 21: I suggest to replace "both debris and no debris flow events" with "both DF and noDF events"

Response

We revised according to reviewer's suggestion.

**1.13** Page 8, Line 1: The acronym "PFDF" does not exist. I suggest to introduce PFDF in the previous part (Page7, Line 7)

Response

We revised according to reviewer's suggestion.

**1.14** Page 8, Line 12, 14, 15: I suggest to replace "$X_1$, $X_2$, $X_3$" with "$X_1$, $X_2$, $X_3$".

Response

We revised according to reviewer's suggestion.

**1.15** Page 10, Line 18, 22: I suggest to replace "NO DF" with "noDF".

Response

We revised according to reviewer's suggestion.

**1.16** Page 11, Line 7: I suggest to replace "NO DF" with "noDF".

Response

We revised according to reviewer's suggestion.

**1.17** Page 19, Table 1: I suggest to replace "(DF=1,NO DF=0)" with "(DF=1, noDF=0)"

Response

We revised according to reviewer's suggestion.

**1.18** Page 22, Table 1: I suggest to replace "(NO Debris Flow)" with "no Debris Flow" and "(No Debris Flow)" with "no Debris Flow".

Response

We revised according to reviewer's suggestion.

**1.19** Page 26, Figure 1: Please show better in the Figure the study area, the four different regions, the location of the DF, and the area of fire-affected catchments. In addition, insert horizontal scale and North indicator symbol.

Response

We have revised Figure 1 following reviewer's suggestions. Specifically, the figure is further zoomed over the regions of interest and the boundaries of the different regions are now outlined to improve visibility. Horizontal scale and North symbol was added. However, it is not possible to show the area of fire affected catchments. The area of fire affected catchments is in the majority less than 1km$^2$ and is not possible to show it on the same figure for all catchments involved. Please note that the purpose of Figure 1 is to demonstrate the geographic extent of the database used and also provide an overview of the location of the events in the database.

**1.20** Page 27, Figure 2: In the blue (AZ region) and black (CO region) boxplots the median is not well visible. Please change the colours using other with major contrast. In addition I suggest to replace in the caption of the Figure "Figure 2 Boxplot for storm duration, storm accumulation and peak 15-min storm accumulation" with "Figure 2 Boxplot for (a) storm duration, (b) storm accumulation and (c) peak 15-min storm accumulation".

Response

We have revised caption and figure according to reviewer's suggestion. We changed the color of the horizontal line corresponding to median values to white, to make it visible for the blue and black boxplot.

**1.21** Page 28, Figure 3: I suggest to replace the caption with "(a) Total rainfall accumulation vs duration and (b) peak 15 minute rainfall accumulation vs duration for Arizona (AZ), California (CA), Colorado (CO) and New Mexico (NM). Colored dots and x symbols correspond to DF and noDF occurrence respectively".

Response

We revised according to reviewer's suggestion.

**1.22** Page 30, Figure 5: I suggest to use a different type of the line (solid line) to represent the full range of variation. The dashed-line produce confusing.

Response

We revised according to reviewer's suggestion.

**Reviewer #2**

**2.1** This paper is intended to present an improved model for forecasting debris-flow occurrence in recently burned areas. The presented material relies exclusively upon previously published data from other sources, and existing methodology. Given the relatively narrow target audience and lack of uniqueness of this topic, my criteria for recommending the publication of this type of study are as follows: 1) New insight into the mechanisms responsible for postwildfire debris-flow occurrence must be gained. 2) Unique methodology for improved predictions must be presented. 3) The methods used to compare model predictions must be defensible, and account for uncertainty in model predictions.

Response

We would like to thank the reviewer for providing his/her comments. Below we provide our detailed response to the points raised.

**2.2** While this paper is well-written, significant revision is required in order to meet the specified criteria for the following reasons:

1) No new insight into the prediction of postwildfire debris-flow occurrence or the pro- cesses responsible for debris-flow generation have been presented within.

Response

We respectfully disagree with the reviewer's opinion that "no new insight into the prediction of post-fire debris flow occurrence" have been presented in this work. We have presented a systematic evaluation of state-of-the-art modeling procedures (two of the models developed in this work) and demonstrated the relationship of their predictive performance with data requirements (sample size and variables considered). Furthermore, in the revised version of the manuscript and per this reviewer's suggestion, we present a comparison of the predicted DF probabilities from LR and RF-based models that allow us to demonstrate with more clarity the superiority of the proposed RF-based vs LR model (currently used by USGS in West US). Therefore we believe that the findings of this work provide collectively new information regarding the approaches for predicting post-fire debris flow occurrence.

It is true that no new insight is presented regarding the understanding of physical processes. We agree that better understanding of the processes responsible for debris flow generation is very important and that it is required to ultimately improve predictive modeling, however process understanding would require a vastly different research approach that is out of the scope of this work.

**2.3** The authors used an existing database of debris-flow occurrence, storm characteristics, and a select number of variables to test an existing modeling approach.

Response

Yes, we relied upon an existing USGS database, which was also used for the development of the LR model in Staley et al. (2017), for the following reasons:

i) It is a database with a large number of events and information associated to burned areas. To the best of our knowledge a similar database that is publicly available was not found for west US.

ii) Given that one of the main objectives of the work was to compare against the current state-of-art model for post-fire DF prediction in west US (developed based on this database), it is appropriate to systematically evaluate the different models (proposed and existing) using the same dataset.

**2.4** The authors did not attempt to incorporate different metrics of topography, fire severity, or other physio- graphic properties of watersheds into this analysis.

Response

During the development of RF-based models we tested a number of combinations of existing variables. In the manuscript, we presented results for the "simple" RF-ED and best performing RF-all model. We did not attempt to generate new parameters (based on topography, fire severity etc.) for the watersheds. This is a significant effort that although could potentially prove beneficial, should be considered as a future step and most likely in collaboration with USGS to get access to the original dataset used to derive the parameters in database (e.g. GIS files of burned areas, satellite images before/after fire etc.).

We believe that the analysis based on the existing dataset still has merit and findings provide new information with important implication for future advances of post-fire DF prediction not only in the west US but elsewhere as well.

We revised conclusions to mention that investigation of new parameters should be considered as a future step:

*"Future work will be focused on examining the model performance using alternative sources of rainfall information (e.g. from weather radar, satellite-based sensors and numerical weather prediction models) and further investigating how extra physiographic parameters (not included in existing database) can potentially improve predictive ability of models."*

**2.5** As such, the author's solely relied upon someone else's calculation of metrics and interpretations of the variables that are important for inclusion in this analysis. Hence, we did not learn anything about new factors or combinations of variables that may be relevant for the prediction of postwildfire debris-flow occurrence.

Response

We would like to note again that a direct and fair comparison of the proposed RF-based models to the currently used procedure for post-fire DF prediction in west US, required us to adopt fully the specificities of the currently used LR model. Otherwise, comparison would not be possible or consistent.

We believe that results from the analysis provide new information that can be useful for the development of predictive models for post-fire DF. Specifically, results from this work provide insight on the relative importance of data requirements (both in terms of size and variables) and model complexity for predicting DF occurrence. Additionally, the revised version of the manuscript provides a more detailed comparison between LR and RF-based models highlighting a) the issues to be considered for further development of LR-based procedures (if possible) and b) an alternative way forward in DF prediction with RF-based models quantifying also their corresponding significance in performance improvement.

**2.6** 2) The authors present a method that utilizes an existing methodology (random forest analysis). It's unclear to me how or why this is significantly different, or a significant advancement, over the machine learning method presented by Kent et al., 2017. It seems that the comparison of model predictions focuses almost exclusively on those between the models presented by the author and the USGS logistic regression model.

Response

Random forest algorithm is an existing algorithm that was used to develop the models presented in this work RF-ED and RF-all. These specific models have not been previously presented. As we argue in the conclusions section, there may very well be other machine learning algorithms that can potentially generate even further improved results. To be able to investigate this, one needs to carry out a systematic evaluation for all the possible models, which is not within the scope of this work. The models evaluated in this work were selected based on a very specific context. First, the ED model is the most widely used threshold for rainfall-induced mass movement phenomena (landslides, debris flows etc) and at the same time the simplest in terms of data requirements and development. The LR model was chosen because it is the current state-of-art and the RF-based models were developed to demonstrate the potential advancements that can be achieved by considering machine learning techniques for post-fire debris flow prediction.

In addition to this, the work presents aspects of the performance evaluation relative to data requirements (sample size and number of variables considered) and in the revised version of the manuscript, it further explores in more detail the differences in DF probability prediction between LR and RF models. Such an evaluation, contrasting in particular the currently operational model in west US does not exist in previous studies.

**2.7** 3) The manuscript provides methodology for a binary classifier model used to predict the occurrence of a postwildfire debris flow. The authors compare their predictions to the USGS logistic regression model, but significantly simplify the USGS model predictions. Logistic regression analysis produces a continuous estimate of the likelihood of debris flow occurrence with values that range from 0 - 1, with 1 representing the highest likelihood. In this study, the authors consider the LR model as a binary classifier model, failing to consider that the model is not inherently binary. The analysis then proceeds to use ROC metrics for binary classifier models to compare the results. The Staley et al 2017 manuscript that the authors draw their comparative statistics from does use ROC metrics to compare findings for a portion of the results and discussion. In my opinion, this is a significant flaw in the Staley et al. 2017 manuscript.

By only viewing the USGS model as a binary classifier, the authors oversimplify the predictions made by the USGS logistic regression model by not accounting for the implicit uncertainty in the logistic regression model predictions. For example, the USGS model might estimate a 0.51 likelihood for a debris flow (i.e. just slightly better than a coin toss), but no debris flow might have occurred. This manuscript considers this to be a failed prediction, but is it really? I would argue that a model that estimated a 0.49 likelihood of not having a debris flow is not incorrect. Hence, the conclusion that the presented model significantly outperforms the USGS model predictions based upon the analysis of the model as a binary classifier is misleading.

Response

We would like to thank the reviewer for providing a thorough discussion on this issue.

Indeed, the example mentioned by the reviewer was one of the reasons that we allowed the probability threshold (used for binary classification) to be set through optimizing TS instead of assigning it a fixed value (e.g. 0.5). This means that if we were dealing with a systematic difference between the LR and RF predicted probabilities, the use of a dynamic probability threshold would take care of this effect and we would not notice a difference in the binary classification performance.

However, we agree with the reviewer that a more thorough evaluation of the DF probabilities predicted by LR and RF-based models would benefit the analysis and strengthen the finding of

this work. For this reason, we carried out extra analysis that investigates this aspect. In the revised version of the manuscript we have added a new section (5.3) that presents and discusses results from the comparison between the distribution of the predicted DF probabilities for the different models (LR, RF-ED and RF-all).

As it is shown in the results of the new Figure 7, the superiority of RF-based models in better separating probabilities associated to DF and noDF events is evident. The LR model suffers from considerable underestimation of the probabilities associated with DF events and overestimation of those associated to noDF events, which creates the issue of "overlap" and the difficulty in accurately identifying between the DF and noDF events.

[revised manuscript text omitted]

---

## Author Response (AR2)

**Authors response to comments by the Editor**

Manuscript reference number: nhess-2018-85

Manuscript title: Evaluation of predictive models for post-fire debris flows occurrence in the western United States.

We would like to thank the editor for his comments. We have revised the manuscript according to these comments and below we provide our detailed response. Comments from the editor are in black font and our response in blue font.

1. We have accepted all suggestions for corrections in text and additions in references etc. Please see the revised manuscript submitted.

2. Page 9: *"From all the different combinations of variables tested (results not shown)…"* Unclear what you mean here. For what reasons these results are not shown? There is really the need to have this? Please clarify

Response
We removed "(results not shown)" to improve clarity.

3. Page 17: *"…we did not account for uncertainty in input parameters"* Could you please add a few words to explain the reasons for this?

Response

We have modified/expand corresponding text, as follows, to further clarify this point.

*"To investigate the impact of input parameter uncertainty we need to first statistically characterize and quantify uncertainty of each input source and then propagate the various uncertainties through the predictive models and evaluate uncertainty in the final predictions. This goes beyond the scope of current work and thus will be a topic of future research."*